# How migrants' transcultural perceptions shape their children's bilingual language development: Insights from a cross-sectional multicultural study

Caroline Barry[1☯], Charlène Mafuta[1☯], Hawa Camara[1,2,3], Bruno Falissard[1], Muriel Bossuroy[3,4,5], Marie Rose Moro[1,2,3,5*], Amalini Simon[1,2,3,4], Dalila Rezzoug[1,2,4,6]

**1** Team DevPsy, Paris-Saclay University, UVSQ, Inserm (National Institute of Health and Medical Research) U 1018, CESP, Villejuif, France, **2** AP-HP (Greater Paris University Hospitals), Paris, France, **3** Cochin Hospital (Maison de Solenn), AP-HP, Paris, France, **4** Avicenne Hospital, Bobigny, AP-HP, France, **5** University of Paris Cité, PCPP, Boulogne-Billancourt, France, **6** Sorbonne Paris Nord University, Villetaneuse, France

☯ These authors contributed equally to this work.
\* marie-rose.moro@aphp.fr

## Abstract

### Background

Little is known about the factors affecting children's language acquisition in transcultural situations and how clinicians can take these children's specific needs into account.

### Objectives

To better understand the acquisition of bilingualism by migrant parents' children, our aim was to study the relations between parental transcultural perceptions and their children's language skills in both the heritage language and the host country's majority language.

### Methods

This cross-sectional study included 114 kindergarten children, born in France to migrant parents speaking Arabic, Tamil, or Soninke. Children's expressive language and comprehension skills were assessed with the ELAL and the N-EEL scales. In semistructured interviews, parents answered questions about perceptions of migration-related changes, extended-family relationships, and transgenerational transmission. Quantitizing methods and regression models were used to assess these factors' potential associations with children's language skills after adjustment for background characteristics and languages used at home.

**Data availability statement:** The dataset underlying this article contains sensitive health and migration-related information, which is considered highly protected under French regulations. For this reason, the data cannot be made publicly available. We have now arranged with the competent authorities that access to de-identified individual-level numerical data can be granted upon request, subject to approval by the relevant ethics and scientific committees and after the signature of a data transfer agreement. Requests should be addressed to the Clinical Research Unit (URC) at Fernand WIDAL Hospital. Contact: Lariboisière site: michele. agor@aphp.fr.

**Funding:** The study was financed by the Direction Générale de l'offre de Soins (DGOS), PHRC 2008 hors cancer The funding enabled the study to be carried out. URL : Organization of the Direction Générale de l'Offre de Soins (DGOS) - Ministry of Labor, Health, Solidarity and the Family (https://sante.gouv.fr/IMG/pdf/resultats_PHRC_2008_hors_cancer.pdf) The DGOS has played no role in the study design, data collection and analysis, decision to publish, or preparation of the manuscript.

**Competing interests:** The authors have declared that no competing interests exist.

**Abbreviations:** ELAL, evaluation langagière pour allophones et primoarrivants (Language Evaluation for Allophones and New Immigrants); N-EEL, new test for language assessment in French; HL, heritage language; ML, majority language (French in this context).

## Results

Children of parents with a strongly positive perception of transgenerational transmission had better expressive skills in their heritage language. However, strongly positive parental perceptions of extended-family relationships and of migration-related changes were independently associated with some poorer skills in the heritage language. None of these transcultural/familial factors was significantly associated with any of the majority language skills assessed.

## Conclusion

This research suggests that parental perceptions of migration, extended-family relationships, and transgenerational transmission are closely related to their children's heritage language skills regardless of the choice of languages spoken at home. Further research on transcultural factors is necessary to illuminate the mechanisms underlying bilingual learning and inform evidence-based practices for clinicians.

## Introduction

Migration flows cause increasing numbers of children to grow up exposed to two languages, their family's heritage language (HL) and their host country's majority language (ML). Bilingual children in immigrant families are exposed to diverse language-learning environments and develop widely varying levels of language skills, often with uneven proficiency between their languages [1,2]. However, acquiring both the heritage and the majority language (i.e., becoming bilingual) is a core component of immigrant children's adaptive development with many ensuing benefits. HL proficiency has been associated with harmonious family relationships, long-term emotional well-being, stronger identity construction, and psychological adjustment [3–6]. Moreover, combined proficiency in both languages has been linked to abilities that foster academic achievement and support social navigation of young people's interpersonal worlds [7].

Navigating multiple languages may be demanding. When bilingual developmental difficulties arise, they pose complex challenges for health professionals, educators, and families alike. While numerous standardized tools are available to assess impairment and guide interventions in monolingual contexts, the situation is more complex in cross-cultural and bilingual contexts. The wide variability of language acquisition makes it difficult to distinguish language disorders from language differences, and additional migration-related risks such as discrimination and acculturative stress may further compound the challenges [8,9]. As highlighted by Stow and Dodd, equitable clinical services require culturally and linguistically appropriate tools and practices to prevent intervention inequities [10]. Identifying factors that may favor or impede bilingual development is therefore essential to help educators, policymakers and health professionals address the specific needs of migrant bilingual children, and optimize interventions to support their language development [11,12].

Sources of variability in bilingual development can be modeled as the result of dynamic interactions between children's internal resources (e.g., cognitive capacities) and external environmental influences [13]. Here, we focus on family environmental factors — a key variability [14]. One highly robust finding of research on early bilingual development is the relation between the quantity of children's exposures to each language (cumulative exposure and current use at home) and their levels of language development (for an overview see [1,15,16]). Nonetheless, the relations between quantitative exposure to languages at home and bilingual development appear complex and may differ according to its role as input/output (heard/spoken) [17,18], the child's receptive and expressive skills (e.g., [19,20]), and in the HL versus the ML [21–23]. A growing body of research is now examining the impact of qualitative aspects of language environment including language richness (e.g., lexical diversity) [24], home literacy practices (e.g., reading to children) [1], and the diversity of interlocutors [25] in each language (e.g., sibling interactions [26–29]).

Areas requiring further investigation include distal environmental factors that may quantitatively and qualitatively shape home exposure to HL/ML [1]. Factors already identified range from family socioeconomic status [24] to family attitudes about languages [1]. Several qualitative studies among parents also point to cultural identity, familism ("the ideological value that cultures assign to familial unity and loyalty" [30]), and emotions [31] as intertwined factors favoring HL development (e.g., [30,32,33]) and emphasize the symbolic function of language transmission for transmitting culture and family heritage. In migrant families, where members often face cultural distance, reshaped family ties, and acculturative stress, addressing children's languages implicitly entails addressing the cultural and intergenerational bonds. Parental attitudes and perceptions toward ethno-cultural identities and migration may therefore be of crucial importance to better understand the development as well as risk and protection processes [4,7]. Some recent studies have begun to explore these issues, examining for example reasons for migration, parental acculturation, home language use, and children's language outcomes, but findings currently remain limited [12].

Our research is grounded in the transcultural clinical approach, the theoretical and methodological foundations of which were built on Devereux's complementarist paradigm, integrating complementary frames of reference, historically, anthropology and psychoanalysis [34,35]. By transcultural, we refer to the dynamic process of cultural transformation with migration, which combines the loss of existing cultural elements, the borrowing and reappropriation of external cultural traits, and the creation of new cultural forms emerging from cultural interaction [36]. This framework has been further operationalized in transcultural psychotherapy (a second-line method drawing on systemic and psychoanalytic family therapy, narrative therapy, and cultural mediation) that has been developed for migrant patients when standard care proves insufficient [37]. Building on this tradition, our clinical work on language and learning disorders with second-generation migrant children has extended the complementarist paradigm. This work suggests that bilingual acquisition must be understood ecologically—not only from a cognitive viewpoint but also from traditional psychological approaches (parents' subjective experiences) and socio-cultural perspectives (anthropology, sociolinguistics, etc.) [38]. This original approach resonates to a certain extent with the theoretical underpinnings of several ecological and integrative models, such as the Integrative Risk and Resilience Model of Adaptation in Immigrant-origin Children [39,40] or the concept of familism [30].

Within this framework, the term heritage language (HL) is today conceptualized as a nondominant language in a host society that holds personal or cultural significance for the individual (e.g., through family or historical ties), regardless of actual proficiency [41]. Unlike alternative labels such as *L1* or *mother tongue*, the concept of HL shifts the focus from linguistic dimension to a symbolic one; HLs are conceptualized as embodiments of community traditions and identities, with each HL a critical medium for transmitting collective memories, values, and cultural practices [42]. The concept of heritage explicitly embeds an intergenerational perspective, linking language to family transmission across time. Pragmatically, HL proficiency may be necessary to sustain communication with relatives, particularly those grandparents who do not master their grandchildren's ML. Relationships with extended family members also broaden the contexts in which the HL is used, enriching children's HL environment both quantitatively and qualitatively. Beyond these practical reasons, qualitative studies have also shed light on the HL's emotional aspects. For instance, some authors, finding that families express a strong

emotional connection to the HL, have proposed that HL is valued for its symbolic maintenance of family bonds and cultural identity [43]. Affective ties with extended kin [30] may lead migrants to make a significant effort to provide crucial support for HL use and for its transmission to their children.

We sought to estimate the relations of three transcultural factors to children's expressive and receptive language skills in both the HL and ML. The factors investigated here were parental perceptions of migration-related changes, transgenerational transmission, and their extended-family relationships. A secondary aim was to determine whether these transcultural factors accounted for language outcomes beyond their effects shaping the choice of language used at home (i.e., child input/output). These results would improve our understanding of what determines children's language pathways in transcultural situations and of how to take their specific needs into account.

## Materials and methods

### Study design

This study used a cross-sectional design combining standardized language assessments with an innovative approach to capture parental transcultural perceptions. We drew on procedures inspired by mixed-methods designs. This two-step process involved conducting semi-structured interviews with parents and then quantitizing their narrative responses (i.e., transforming them into numerical scores [44]), thereby generating reproducible variables for statistical analysis. The similar strategies applied in health and social sciences to quantify familial relationships [45] inspired the approach used here. The rationale for this approach was twofold. First, interviews allow parents to voice their feelings and opinions, providing access to their subjective experiences of complex transcultural domains in a more open-ended manner than would be possible with a standardized questionnaire; moreover, when we began this study, no standardized questionnaire existed on this subject, nor has any been published since. Second, a quantitative step is required to test associations between transcultural parental factors and children's receptive and expressive skills in HL and ML. By grounding the construction of quantitative variables in qualitative parental accounts, our design aimed to operationalize complex transcultural constructs and to explore their relationships with children's bilingual development in ways that reflect the complexity of their language pathways in migration. This strategy aligned with our theoretical framework of complementary perspectives and allowed us to capitalize on the strengths of each tradition—qualitative data offering depth and contextual richness, and quantitative modeling providing statistical robustness and generalizable estimates—to enhance the interpretability of our findings.

### Study population, and data collection

This study was part of the cross-sectional study titled "Validation of Avicenne's ELAL: Instrument of mother tongue assessment and impact of familial and transcultural factors on early bilingualism in migrants' children born in France." Families were recruited from preschools, preventive medical care centers, and cultural associations in Paris and its surrounding area. Eligible families included at least one first- or second-generation migrant parent who spoke Arabic, Tamil, or Soninke, along with at least one of their children aged 3.5 to 6.5 years, born in France, and attending either the second or third year of preschool. Families in which each parent spoke a different HL were excluded. Only one dyad (a migrant parent and the relevant child) was recruited per family. Recruitment occurred between June 30, 2011, and July 22, 2014.

The rationale for the selection of this population was to maximize cultural diversity through a strategic yet pragmatic selection of three contexts, focusing on prevalent migrant groups in the region spanning three distinct linguistic families and migration histories. These groups reflect major migration streams to France—North Africa and the Middle East, South Asia, and Sub-Saharan West Africa—whose languages belong to three genealogically unrelated families: Semitic (Maghrebi Arabic), Dravidian (Tamil), and Niger-Congo (Soninke). Maghrebi Arabic families are embedded in a post-colonial labor migration context, with parents typically bilingual but facing stigmatization. Tamil families represent a recent migration linked to the Sri Lankan civil conflict, sustained by strong community practices of language and cultural

transmission. Soninke families, largely from West Africa, migrated for economic reasons; Soninke is primarily oral and less institutionalized in written form: it is maintained in France through solidarity networks and ritual events. While neither an exhaustive nor a representative sample of all migrant populations, these groups together constitute a clinically relevant sample whose heterogeneity strengthens the ecological and analytical validity of the study.

Data collection involved an assessment of the children's language skills in both languages, conducted by a researcher-interpreter pair, as well as an interview with the parents to gather information on family language practices and transcultural factors. Written informed consent was obtained from all participating parents on behalf of themselves and their children. Sponsored by Assistance Publique – Hôpitaux de Paris (AP-HP), the study received approval from the ethics committee of the University Hospitals of North Paris (CEERB-N°IRB 0006477, April 8, 2011). The primary study initially included 145 parent-child dyads, but 31 dyads were excluded from this analysis because no parent completed the transcultural interview.

## Measures

### Children's language outcomes

HL (Arabic, Tamil, or Soninke) acquisition was measured with the ELAL scale (Language Evaluation for Allophones and New Immigrants, available at https://www.webelal.org) [46,47], an instrument designed to assess children's skills in any minority language. It has three subscales: Comprehension (lexical-semantic vocabulary and morphosyntax); Expression (lexical-semantic vocabulary); and Storytelling (morphosyntactic and narrative skills). The higher the comprehension and expression scores, the better the skills. The Storytelling scale discriminated children with isolated vocabulary expression skills from children able to formulate a story (Yes/No).

ML (French) acquisition was assessed with the New Test for Language Assessment (Nouvelles Epreuves pour l'Examen du Language, N-EEL), a validated reference scale assessing comprehension and expression in French [48]. We selected the N-EEL subscales assessing skills equivalent to those tested by the ELAL scale: N-EEL items assess comprehension (lexical-semantic vocabulary, morphosyntax) and expression (lexical-semantic vocabulary).

Thus, the study examined five language outcomes to assess this issue: ML comprehension and expression skills, HL comprehension and production skills, and ability to formulate a narrative in the HL. To compare HL and ML outcomes, each ELAL and N-EEL score was rescaled to a mean of zero and a standard deviation of one.

### Factors related to parents' transcultural perceptions

Here, perception designates a mental process encompassing experiences, emotions, and allegiances. The instrument we developed has two components: the first a semistructured interview guide for the investigators who interviewed parents about transcultural/familial factors; and the second a Likert-type numerical value scoring grid for each factor, to be used by the raters to quantitize the qualitative corpus made up of parents' narrative answers. Supplementary S1 File presents the tool development process and testing phases, while S2 File provides excerpts (translated into English) from the semi-structured interview guide used by the investigators to interview parents about transcultural factors, as well the Likert-type numerical scoring grids used by raters to quantitize the qualitative corpus for each factor.

The transcultural factors we investigated here are detailed below, with examples of the open-ended interview questions.

1) Parental perception of migration-related change: We asked parents about their perceptions of life changes accompanying their migration to live in France. The starting question was: "What do you think of the changes in your life since you arrived in France?"

2) Parental perception of extended-family relationships: We sought to evaluate each interviewee's perceptions of their relationships with members of their extended (not nuclear) family, in France or elsewhere, and independently of

frequency of contact. Core questions to parents were: How are things with your family: parents, grandparents, siblings, aunts and uncles, cousins etc.? Can you tell me about your relationship with your family?

3) Parental perception of transgenerational transmission: We explored the parents' perceptions of transmitting their family's culture (in the broad sense) and heritage, that is, how much the parents wish to place their child into their family line. Items questioned whether parents have transmitted elements of their family history to the child and what elements if any the parent have identified in the child that remind them of their own parents or of themselves as a child (What did your child take from his or her maternal or paternal grandparents: physical resemblance, character, values...?).

In practice, semistructured interviews were conducted in the language chosen by the parents, audio-recorded, translated (when necessary), and then fully transcribed. Next two researchers (not the same pair of raters for all subjects) independently used the corresponding scoring grids to rate the intensity of each factor for each subject. Value range scores obtained by quantization had direction and magnitude representing the intensity of the factor (ranging from −5–5): the more positively the parent perceived a factor, the higher the score. For each variable, anchor points were operationalized by a specific statement that best exemplified that specific level of intensity. For example, concerning the factor of transgenerational transmission, the statement descriptive of a rating of −5 is: "*the subject is at odds with transmission, which may be rejected or represent an obstacle for the subject. The question of transmission can be associated with insecurity and abandonment. Transgenerational transmission is not encouraged and is blocked.*" The statement descriptive of a rating of 5 (positive) was "the participant values this transmission, writing him- or herself and the child in a filiation; cultural transmission appears direct and intense, with strong intention".

These tools were administered by 7 professionals: 5 clinical psychologists with 10 years of specialized experience in individual, group, and transcultural therapy; a psycholinguist specialized in bilingualism and language assessment with 20 years of experience; and a child/adolescent psychiatrist with 15 years of specialization in language assessment in bilingual contexts and in transcultural work with migrant children and their families. Two raters independently applied the scoring for each factor, initially blinded to the other rater's score. The intraclass correlation (ICC), that is, the interrater agreement measured with random effects model), was 0.65 (95%CI 0.53–0.75) for transgenerational transmission, 0.77 (95%CI, 0.69–0.83) for extended-family relationships, and 0.84 (95%CI 0.77–0.89) for migration-related change. Then the two discussed and recalibrated each score in case of divergence. When discrepancies could not be resolved, the data were reanalyzed by the entire expert group. The final score was the consensus obtained.

Finally, scores were discretized by using a binary split at the median of the distribution. Parents who obtained a score above the median will be referred as having a "strongly positive" perception. Reducing the continuous range of scores to a binary variable allowed us to avoid assumptions about the linearity of the relationships between parents' transcultural perceptions and children's language outcomes, thereby providing a more robust analysis of the data. Clinically, we hypothesized that perceptions above the median level might have a particular impact on the outcomes. For the perception of the migration-related change variable, a third-level category was created to consider those who arrived very young in France or second-generation parents born in France; they were not asked the question because they had no memories from before the migration.

## Language use and sociodemographic data

Language use at home was assessed with the following questions: What language (or languages) do you use to speak to your child? What language does your child use to speak to you? And to speak with his/her siblings? The potential answers were: HL, ML, or both.

The following sociodemographic covariates were also considered in the analyses since previous research has noted these to be sources of individual differences in bilingual exposure: the child's age [49] and sex [13], status as a first-born or only child [29], length of host country residency by the parent participating in the study [29], and parents' (highest) occupational category [24].

## Statistical methods

All statistical analyses were performed with R version 4.0.0. [50] Two-sided *P* values < .05 were considered statistically significant.

Descriptive statistics were provided. Linear and logistic multiple regression models were then used to identify transcultural variables significantly associated with language outcomes (one model per outcome). To examine the extent to which the selected covariates explained the association between parents' transcultural perceptions and child language outcomes, we fitted four models by outcome with incremental adjustment levels. Variables included in the adjustment were guided by a review of the literature and theoretical considerations of their potential as confounders. The initial models (Model M0, one per outcome) included the three transcultural factors and the child's age as explanatory variables. Next, three additional models were fitted for each outcome, successively adjusting for children's characteristics (M1), parental socio-demographic characteristics (M2), and languages used at home (M3).

To take the sparse covariate missing data into account (less than 2% of values), we used simple imputation methods (medians replaced missing numeric data and modes replaced missing categorical variables.

## Results

### Sample characteristics

This study included 114 parent-child dyads, 34 with Maghrebi Arabic (30%), 44 Tamil (39%) and 36 Soninke (32%) (percentages do not sum to 100 due to rounding) as their HL. The participating parents had lived in France for 14.6 years (SD 8) on average, and the children's mean age was 5.4 years (SD 0.7). Table 1 details the dyads' characteristics. Parents and children differed substantially in their active language use at home. Almost all parents (97%) used HL (43% exclusively and 54% both) to speak to their children, while 59% of the children responded with the HL (17% exclusively and 42% with both). An even smaller proportion of children used the HL with siblings (among the 110 with siblings 9% exclusively and 33% both). Finally, 69 (61%) children had sufficient narrative skills to formulate a story in their HL.

### Distribution of the scores of transcultural factors

The parents reported a mostly positive perception of migration-related changes (median 3.0, range −4–5), extended-family relationships (median 4.2, range −3–5) as well as of transgenerational transmission (median 3.8, range −1–5). Table 2 presents descriptive statistics for HL use in families by level of parental perception of transcultural domains. Minimal variation was observed in parents' own HL use, which remained high across all subgroups. However, the children's HL use appeared more variable. Notably, the decrease in HL usage between parents and children tended to be particularly marked in families with a parent strongly invested in extended-family relationships or with strongly positive perceptions of migration-related changes.

### Associations between language skills and transcultural factors

Fig 1 presents the regression model results across all outcomes (comprehension, expression, narrative skills, one per column) and each adjustment (M0 to M3, one per row), while Supplementary Table S1 reports all numeric results of the M3 (i.e., fully adjusted) models. As Fig 1 shows, the direction of estimates is consistent across all HL outcomes (comprehension, expression, narrative skills) and all adjustments. Children whose parent's perception of transgenerational transmission was strongly positive had higher scores on the ELAL expression subscale (Md0 beta +0.50 95%CI [0.16, 0.83, p = 0.0042) than children whose parent's perception of transgenerational transmission was poorer. Characteristics of children or families accounted for about one-third of this association and thus explained it only partially (beta perception of transgenerational transmission +0.32 95%CI [0.06, 0.59, p = 0.0180 in model Md2). After adjusting for language use, we still found that children with a parent with a strongly positive perception of transgenerational transmission had higher ELAL

**Table 1. Characteristics of the sample (n = 114).**

| Dyad Characteristics | | n (%) |
|---|---|---|
| **Child's gender** | | |
| | Male | 51 (45%) |
| | Female | 63 (55%) |
| **Sibling rank** | | |
| | First child (Oldest) | 41 (36%) |
| | Rank >= 2 | 73 (64%) |
| **Parents** | | |
| | Father | 8 (7%) |
| | Mother | 106 (93%) |
| **Parents' highest occupational category** | | |
| | Worker | 14 (13%) |
| | Employee | 69 (63%) |
| | Other [a] | 26 (24%) |
| **Parent's migration status** | | |
| | First-generation immigrant | 97 (85%) |
| | Second-generation immigrant | 17 (15%) |
| **Language spoken by the parent to the child at home** | | |
| | Majority language only | 4 (4%) |
| | Heritage language only | 49 (43%) |
| | Both languages | 61 (54%) |
| **Language spoken by the child to the parent at home** | | |
| | Majority language only | 47 (41%) |
| | Heritage language only | 19 (17%) |
| | Both languages | 48 (42%) |
| **Language spoken by the child to the siblings at home[b]** | | |
| | Majority language only | 63 (58%) |
| | Heritage language only | 10 (9%) |
| | Both languages | 36 (33%) |
| **Perception of migration-related change** | | |
| | Strongly positive perception | 53 (48%) |
| | Less favorable perception | 38 (35%) |
| | Not concerned [c] | 19 (17%) |
| **Perception of transgenerational transmission** | | |
| | Strongly positive perception | 66 (58%) |
| | Less favorable perception | 48 (42%) |
| **Perception of extended-family relationship** | | |
| | Strongly positive perception | 65 (57%) |
| | Less favorable perception | 49 (43%) |

All variables are described as counts (%).

[a]: craftsman, business manager, executive, higher intellectual profession, intermediate profession: https://www.insee.fr/en/metadonnees/definition/c1493

[b] Among the 110 who have siblings

[c] Second-generation migrants or first-generation migrants who arrived in France before the age of 3 years

**Table 2. Descriptive statistics for HL use in families by level of parental perception of transcultural domains.**

| | MRC0 (n=19) | MRC+ (n=53) | MRC- (n=38) | TGT+ (n=66) | TGT- (n=48) | EFR+ (n=65) | EFR- (n=49) |
|---|---|---|---|---|---|---|---|
| Parent to Child | 89% (69, 98) | 97% (87,100) | 98% (90, 100) | 100 (93, 100) | 94% (85, 98) | 94% (83, 98) | 98% (92, 100) |
| Child to Parent | 42% (23, 64) | 55% (40, 77) | 65% (55, 79) | 62% (48, 75) | 56% (44, 67) | 51% (37, 64) | 65% (52, 75) |
| Child to Siblings | 35% (17, 59) | 28% (16, 44) | 54% (41, 67) | 43% (30, 58) | 41% (30, 54) | 36% (24, 50) | 47% (35, 59) |

All frequencies are described as percentages of parents or children using the HL only or together with the ML when speaking and their 95% confidence intervals.

MRC+ Subgroups of first-generation parents with a strongly positive perception of migration-related change, MRC-: Subgroups of first-generation parents with a less favorable perception of migration-related change, MRC0: No memory before migration, TGT+/TGT-: Subgroups of parents who reported a strongly positive/less favorable perception of transgenerational transmission (TGT±), EFR+/EFR-: Subgroups of parents who reported a strongly positive/less favorable perception of extended-family relationships (EFR).

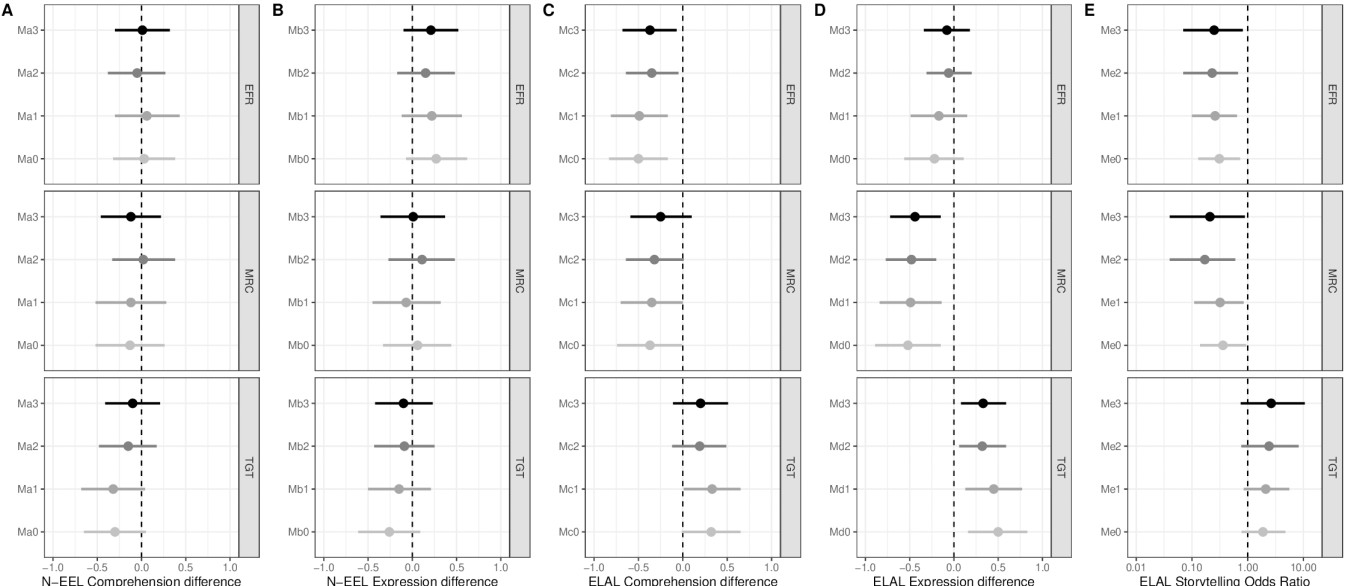

**Fig 1. Associations between parental perceptions of transcultural factors and the children's language outcomes.** Estimates are beta coefficients for linear regression (A, B, C, D) and odds ratios (E) for logistic regressions with 95% confidence intervals.

expression scores (beta +0.33, 95%CI [0.08, 0.59, p=0.0127 in the fully adjusted model Md3). Associations between parental perceptions of transgenerational transmission and the other ELAL subscales did not reach statistical significance (comprehension, Mc3: beta=+0.20, 95% CI [–0.11, 0.51], p=0.2101; narrative, Me3: adjusted odds ratio ORa=2.64, 95% CI [0.75, 10.57], p=0.1446).

Each version of model 0 included the three transcultural factors and the child's age. Outcomes are indicated at the bottom of each part. Each version of model 1 was additionally adjusted for the child's sex, sibling rank (oldest), and the score assessing the corresponding linguistic skill in the non-target language (e.g., N-EEL comprehension scores when the outcome is the ELAL comprehension scores, that is, to assess the association between the transcultural variables and the comprehension skills specifically in the HL, comprehension skills in the other language remaining equal).

Each version of model 2 was additionally adjusted for the number of years lived in France by the parent included in the study, the highest socioprofessional social category of the parents, and the HL (Tamil, Arabic, or Soninke).

Each version of model 3 was additionally adjusted for languages used by the included parent to talk to the child, by the child to talk to the included parent and by the child with siblings.

Children whose parent's perception of migration-related changes was strongly positive had poorer expression skills and lower ELAL expression scores (Md0 beta −0.52 95%CI [−0.89, −0.15], p = 0.0063) than those whose parent's perception was less favorable. This estimate barely fell after adjustment for all covariates including language use at home (Md3 beta −0.44 95%CI [−0.72, −0.15], p = 0.0036). Furthermore, children whose parent's perception of migration-related changes was strongly positive were less likely to have narrative skills (Me0 ORa 0.36 95%CI [0.14, 0.94], p = 0.0385, Me3 ORa 0.21 95%CI [0.04, 0.89], p = 0.0441). The gap in comprehension skills was smaller and not statistically significant after adjustment (Mc0 beta −0.37 95%CI [−0.74, −0.01], p = 0.0457, Mc3 beta −0.25 95%CI [−0.59, 0.10], p = 0.1662).

Children whose parent's perception of extended-family relationships was strongly positive had poorer HL language skills than those whose parent perceived extended-family relationships less favorably, with lower comprehension (Mc0 beta −0.50 95%CI [−0.83, −0.17], p = 0.0036) and narrative skill (Me0 ORa 0.31 95%CI [0.13, 0.73], p = 0.0083) scores. These differences persisted after adjustment for characteristics of both the child and family and for language use (Mc3 beta −0.37 95%CI [−0.68, −0.07], p = 0.0184; Me3 ORa 0.25 95%CI [0.07, 0.82], p = 0.0272).

No statistically significant association was observed between the child's ML skills and any transcultural variable (see Fig 1 and S1 Table).

## Discussion

This cross-sectional study explored how parental transcultural factors were related to both language use at home and language skill acquisition among children of migrant parents in France. Using data from 114 families with three different languages and cultural backgrounds, we found that children whose parents perceived transgenerational transmission as strongly positive had a richer expressive vocabulary than children whose parents perceived transgenerational transmission less positively. However, children of parents with strongly positive perceptions of extended-family relationships or migration-related changes had poorer HL skills than children whose parents perceived these factors less positively.

Neither background sociodemographic characteristics nor the child's language capacities measured by their ML skills confounded these associations. Although these results cannot demonstrate causal mechanisms, differences in the transcultural factors (perceptions, beliefs, investment) may well affect parental HL transmission behaviors, which may in turn shape the child's language exposure either quantitatively or qualitatively and may thus shed light on our findings. From the quantitative perspective, our results overall did not support the assumption that substantial differences in language use at home explain these associations, given that i) neither parental nor child HL use was statistically significantly associated with transgenerational transmission or extended-family relationship factors; ii) a full mediating effect was excluded by regression model results showing that language outcomes for children remained associated with transcultural and familial factors after accounting for HL use at home.

Another potential explanation may be that emotional values attached to languages and perceptions of transcultural factors affect each other [51,52]. Desire for transgenerational transmission of culture and family legacy may thus be significantly involved in HL acquisition. Mothers accounted for 93% of participating parents, and the appreciation by some of the emancipation they experienced in France may help explain the link between some of their perceptions in migration-related changes and their children's language skills. Moreover, assimilationism in France encourages investment in change.

Several hypotheses may shed light on the negative association between positive perceptions of extended-family relationships and children's HL skills. When parents receive strong affective support from their extended family, the symbolic function of HL as the primary link to cultural identity may become less central within the nuclear family, diverting parental investment away from HL and reducing pressure on the child to maintain it. Another related explanation is that extended-family members living in France may interact mainly in French, so that supportive ties increase children's ML

rather than HL exposure. A further possibility is that very positive perceptions of extended-family relationships concern geographically distant relatives—a frequent situation in migration contexts—where distance fosters idealization without providing effective HL interactions. The latter two interpretations point to gaps in our instrument, which captures the valence of family ties but neither the frequency nor the language used in these conversations. Future studies should refine these measures and combine them with in-depth qualitative analyses among families with highly positive perceptions to better understand this counterintuitive result.

Studies of transcultural factors and the development of bilingualism in children are limited, and even fewer measure children's HL skills. Our results can nonetheless be contrasted with previous US cross-sectional studies, particularly among Mexican [12,53] and Chinese [12,53,54] immigrant families. These focused on reasons for migration [12], parental ethnic-cultural orientation [12,54] (e.g., values, cultural practices, interactions with people from their country of origin, or HL proficiency), and perceptions of bilingualism's value for their children [53]. Reasons for migration [12] were not associated with parental ethnic-cultural orientation, language input, or children's HL vocabulary. No significant direct relation was found between parental ethnic-cultural orientation [12,54] and children's HL expressive and receptive proficiency, but path analysis models suggested an indirect relationship mediated by children's language input. Our results on transgenerational transmission are consistent with an article focused on parental perceptions of bilingualism [53], which the authors reported were associated with children's HL proficiency but not significantly with their relative HL/ML input and output at home. Their regression analysis results showed an association between children's HL vocabulary and parental perception of bilingualism after controlling for the child's language use.

Our results about perceptions of extended-family relationships appear to contradict those of qualitative studies (e.g., [30] for Hispanic families in Canada) that suggest familism as a factor favoring HL maintenance. Nonetheless the negative association observed between perceptions of extended-family relationships and HL use agrees with results from a quantitative study [55] that examined associations between family relationships and language preference in a sample of children of Mexican descent. Those who preferred using English (exclusively or with Spanish) reported better family relationships than those who preferred using Spanish.

Our study found no association between parents' transcultural perceptions and children's ML skills, likely because the primary drivers of ML acquisition for schoolchildren are school and community environments, rather than the home.

## Strengths, limitations, and future directions

The study's strengths included a rarely investigated population and the ability to adjust for varied confounders. Quantization of narrative information enabled us to obtain measures of complex transcultural factors, although their construct validity remains to be fully established. The measure developed for this study aims to capture parents' subjective perspectives. It was designed to aid in understanding complex phenomena that cannot be directly addressed. While this approach has limitations due to the challenge of interpreting subjective representations through numeric values, it also offers originality by focusing on individuals in the full complexity of their mental life.

Several aspects of the design also limit the scope of our conclusions. The recruitment of volunteers from a single metropolitan area limits generalizability, and sample size prevents detection of small effects. The cross-sectional design prevents any inference of causality between parents' perceptions of transcultural factors and children's language outcomes. As previously reported [54], children are not passive recipients and their language use may shape their parents' perceptions. Furthermore, this study evaluated children aged 3–6 years, whose language profiles can change rapidly and substantially. Our results might best be viewed as a snapshot of the interrelations between parental perceptions and child language skills in this age group but potentially quite different for toddlers or teens. Longitudinal studies are needed to understand the developmental dynamics of bilingualism. Further research should explore more comprehensive quantitative measures of languages spoken at home and potential interactions with qualitative aspects of language environment, notably literacy practices or the languages of extended-family interlocutors.

## Practical implications

Our results point to several practical implications. First, we suggest that parental transcultural perceptions—how parents value migration-related changes, extended family relationships, and intergenerational transmission—play a role in children's HL acquisition. Extrapolating these results, we propose that in clinical and educational settings, questions about children's language use should not be limited to assessing developmental milestones, but could also serve as entry points into understanding identity formation and family relationships. Systematically documenting such perceptions could help identify children at risk of HL erosion and guide more tailored support. Acknowledging transcultural factors that support language development may help families and educators sustain and validate home-based practices as well as contribute to reducing the stigma often attached to minority languages.

Our findings also have implications for children's health education: supporting HL alongside ML may strengthen health literacy by enabling children to navigate multiple cultural contexts, communicate effectively with both family and healthcare providers, and integrate health knowledge into daily life. Ultimately, rather than viewing multilingualism as a barrier, this approach reframes it as a resource to be nurtured—for the benefit of children, families, and society.

## Conclusions

This study shows that parental transcultural perceptions are significant factors rather than minor variables in bilingual development. Parental perceptions about transgenerational transmission may be a promising target area for supporting children's bilingual development, especially when the parents perceive their new life after migration very positively. Although HL proficiency has been associated with positive health outcomes for these populations, much remains to be learned about its underlying mechanisms. Further research on transcultural factors could inform childcare professionals' evidence-based practices and help pediatricians to improve the care of the growing population of migrant children [56,57].

## Supporting information

**S1 File. Rating grid development, testing, and application processes.**
(DOCX)

**S2 File. Excerpts of the semi-structured interview guide for investigators interviewing parents about transcultural factors, and the Likert-type numeric scoring grids used by raters to quantitize the qualitative corpus for each factor.**
(DOCX)

**S1 Table. Full results of M3 multiple regression models.** Estimates are beta coefficients for linear regression (Models A, B, C, and D) and odds ratios (Model E) for logistic regressions with 95% confidence intervals. [a] Model Ma3 adjusted for ELAL Expression scale score, Mb3 adjusted for ELAL Comprehension scale score, Mc3 adjusted for N-EEL Expression scale score, and Md3 adjusted for N-EEL Comprehension scale score. No such adjustment for Me3 because ML storytelling skills were not scored with the N-EEL scale.
(DOCX)

## Acknowledgments

We thank Dr. Malika Bennabi, Dr. Fatima Touhami, and Jo Ann Cahn for their review of the manuscript. Thanks to Eric Vicaut, Veronique Jouis, and Walid Maklouf of the URC of Fernand Widal Hospital for their regulatory and logistic support on the study.

## Author contributions

**Conceptualization:** Caroline Barry, Charlène Mafuta, Bruno Falissard, Marie Rose Moro, Dalila Rezzoug.

**Data curation:** Caroline Barry, Charlène Mafuta, Hawa Camara, Muriel Bossuroy, Amalini Simon.

**Formal analysis:** Caroline Barry, Charlène Mafuta.

**Funding acquisition:** Marie Rose Moro, Dalila Rezzoug.

**Investigation:** Hawa Camara, Muriel Bossuroy, Amalini Simon, Dalila Rezzoug.

**Methodology:** Caroline Barry, Charlène Mafuta, Hawa Camara, Bruno Falissard, Marie Rose Moro, Dalila Rezzoug.

**Project administration:** Dalila Rezzoug.

**Resources:** Hawa Camara, Muriel Bossuroy, Amalini Simon, Dalila Rezzoug.

**Software:** Caroline Barry, Charlène Mafuta.

**Supervision:** Caroline Barry, Marie Rose Moro, Dalila Rezzoug.

**Validation:** Caroline Barry, Charlène Mafuta.

**Writing – original draft:** Charlène Mafuta.

**Writing – review & editing:** Caroline Barry, Hawa Camara, Bruno Falissard, Muriel Bossuroy, Marie Rose Moro, Amalini Simon, Dalila Rezzoug.

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
