## [Decision Letter · Decision Letter 0]

4 Jun 2025

PONE-D-24-60523How transcultural perceptions of migrants shape the bilingual language development of their children?  Insights from a  multicultural  cross-sectional studyPLOS ONE?

Dear Dr. Moro,

We look forward to receiving your revised manuscript.

Kind regards,

Doris V. Ortega-Altamirano, PhD

Academic Editor

PLOS ONE

Journal Requirements:

Additional Editor Comments:

1) It is a good study, and I suggest improving the manuscript before sending it to other reviewers. 1) to delve into the theories behind the concepts used. Concepts such as inherited language, extended family relationships and transculturality.

2) I suggest to the authors that in their methods they should make explicit the justification of: why is it sufficient to study population with three languages -Arabic, Tamil and Soninke-, is it sufficient to consider them representative of transculturality? In the letter to the editor there is a glimpse of what the explanation might be but it is not in the manuscript.

3) Authors are required to justify the use of measurement instruments and how they were planned to be used with the information collected from the interviews; as well as extending the explanation of the procedure for interpreting numerical data in the light of qualitative data obtained, and vice versa.

4) Authors are required to clarify statements such as: 374-376: “Several hypotheses may shed light on the negative relationship between perceptions of extended-family relationships and HL skills. We cannot exclude possible flaws in measuring perceptions of extended-family relationships or failure to adjust for confounding variables”. The above, how it affects the achievement of the study objective reported by the authors.

Reviewers' comments:

Reviewer's Responses to Questions

**Comments to the Author**

1. Is the manuscript technically sound, and do the data support the conclusions?

Reviewer #1: No

Reviewer #2: Yes

2. Has the statistical analysis been performed appropriately and rigorously?

Reviewer #1: No

Reviewer #2: Yes

3. Have the authors made all data underlying the findings in their manuscript fully available?

Reviewer #1: Yes

Reviewer #2: No

4. Is the manuscript presented in an intelligible fashion and written in standard English?

Reviewer #1: Yes

Reviewer #2: Yes

Reviewer #1: Though timely and relevant, the manuscript suffers from major conceptual and methodological problems that severely restrict its contribution. That is, this study lacks a strong theoretical basis. The key constructs--migration-related changes, extended-family relationships, and transgenerational transmissions--are not defined or theoretically grounded in existing transcultural or socio-linguistic literature. Thus, the framing feels vague and inconsistent throughout.

A quantitative analysis of open-ended interview data raises serious concerns about its validity and reliability. Evidence of psychometric testing or inter-rater agreement is not available. Moreover, the reduction of moods and impressions to binary categories distorts complex cultural experiences. The authors draw strong conclusions from weak associations. Some of the findings are marginally significant or inconsistent, yet interpretable. For instance, the unexpected negative link between positive extended-family perceptions and heritage language ability is scantily explained, appearing rather speculative.

The study used a small, non-random sample from one urban area in France, of which only participants from three language groups could be counted against the accolade for being "multicultural". Cross-sectional design does not allow for any sort of causal inference, even if the authors' wording hints at that. Thus, the authors need to clarify how language proficiency was assessed and ensure that a balance is made in the heritage/majority language mix. The statistical approach would need to be revised, probably better through latent constructs or stronger methodologies overall. Inter-rater reliability should also be reported. Finally, practical implications for clinicians and educators should be further spelled out.

The authors should also clarify how language proficiency was assessed and ensure the balance between heritage and majority languages. The statistical approach should be reconsidered, possibly using latent constructs or more robust techniques. Inter-rater reliability should be reported. Finally, practical implications for clinicians and educators should be made more concrete.

Reviewer #2: Reviewer Comments

Overall Evaluation:

This manuscript presents original research exploring the relationship between parental transcultural perceptions and bilingual language development among children of migrant families in France. The topic is timely, relevant, and grounded in a strong conceptual and methodological framework. The study is well-designed, the analyses are technically sound, and the conclusions are data-driven. However, some areas would benefit from minor clarification and improvement, especially in reporting and framing.

Evaluation by Criteria

1. Originality and Novelty -Meets standard.

The study addresses a novel question by linking transcultural parental perceptions with heritage and majority language development. Its innovative and well-executed approach to " quantifying " qualitative data for regression analysis is also noteworthy.

2. Prior Publication- Meets standard.

The results have not been published elsewhere, as declared in the manuscript.

3.Technical Rigor and Detail- Meets standard.

The methodology—including sample selection, instruments (ELAL and N-EEL), and statistical modeling—is clearly described and appropriate. The use of multiple adjusted regression models strengthens the findings.

Recommendation: Consider elaborating on the construct validity and inter-rater reliability of the scoring system used to "quantitize" parental perceptions, as this is critical to the integrity of the analysis.

4. Conclusions and Data Support- Meets standard.

Conclusions are well-grounded in the data, clearly acknowledging limitations such as the cross-sectional design and lack of causal inference.

5.Clarity and Use of English- Meets standard.

The article is written in clear and intelligible English. Minor copyediting could further improve flow, but no major issues are present.

6.Ethical Standards- Meets standard.

Ethics approval was obtained from an appropriate IRB (North University Hospitals of Paris). Informed consent procedures are properly described.

7.Research Integrity- Meets standard.

The study adheres to good research practices, and there are no signs of data manipulation or undue bias.

8.Reporting Guidelines and Data Availability- Partially meets standard.

The authors note that data cannot be shared due to French privacy laws. While this may be valid, PLOS ONE requires full transparency.

Recommendation: The authors should clarify what aspects of the data are restricted and whether anonymized or partial datasets (e.g., coded scores) could be made available under controlled access or through a trusted repository.

Final Recommendation:

Minor Revision

The manuscript strongly contributes to the literature on bilingualism and migration studies. I recommend acceptance pending minor revision, specifically to clarify the data availability constraints and add more detail regarding the quantification of transcultural factors.

**Do you want your identity to be public for this peer review?** For information about this choice, including consent withdrawal, please see our Privacy Policy

Reviewer #1: No

Reviewer #2: **Yes: ** Manal Hani Ahmad

---

## [Author Response · Author response to Decision Letter 1]

10 Sep 2025

10/09/2025

Dear Dr. Ortega-Altamirano,

Thank you for your constructive comments on our manuscript and for the opportunity to submit a revision.

We attach the clean, revised manuscript and the version with the changes tracked. Our responses to the reviewers’ comments are provided below. We hope that the changes we have made address all your concerns and that manuscript describing innovative research in the field of children's language development in a multicultural society can now be published.

We also note that we have modified the title to make it easier and clearer for readers to understand. If you think it is too late to modify the title, we will understand. We have added the contact details of the clinical research unit that provides access to the data from this research.

Please do not hesitate to contact us if you have any further questions or suggestions.

We look forward to hearing from you at your earliest convenience.

Yours sincerely,

Professor Marie Rose Moro

• Professor at Paris Cité University and Research Director at the National Institute of Medical Research (CESP/INSERM).

• Head of Department at the Maison de Solenn (AP-HP, Hôpital Cochin, Paris, France) : www.maisondesolenn.fr

• Member of the French National Academy of Medicine.

• Member of the Institut Universitaire de France (IUF).

• Honorary doctorate from the University of Mons (Belgium).

• Scientific director of the transcultural journal L’autre (www.revuelautre.com)

• President of the International Association of Ethnopsychiatry (www.aiep-transculturel.com).

Responses to the editor

I suggest to the authors to emphasize on the reasons for addressing the research question.

Response:

Thank you for your question.

This study was initiated in response to clinical needs identified at our hospitals, located in Paris and its multicultural and multilingual suburbs. Multilingualism among children is an issue in many other cities in Europe and around the world. Existing language assessment tools are inappropriate for assessing the language abilities of bilingual migrant children as they often underestimate their skills due to the distributed and nonlinear nature of bilingual development.

To address this issue, our research team has previously developed several tools applying a transcultural methodology tailored to multicultural contexts. One is the Avicenne ELAL, a tool designed to evaluate heritage language skills acquired within the home. We also created a parent interview protocol to contextualize children’s language abilities by gathering data on family language practices and parental attitudes towards cultural and linguistic transmission.

The use of minority languages varies significantly according to family structure (nuclear versus extended), social isolation, and opportunities for return visits to the country of origin. As heritage languages are not consistently passed on in all migrant families, our aim was to explore the parental perceptions and practices associated with better language outcomes in children.

To clarify the rationale behind this research, we have added the following text at the start of the introduction:

Migration flows cause increasing numbers of children to grow up exposed to two languages, their family’s heritage language (HL) and their host country's majority language (ML). Bilingual children in immigrant families are exposed to diverse language-learning environments and develop widely varying levels of language skills, often with uneven proficiency between their languages [1],[2]. However, acquiring both the heritage and the majority language (i.e., becoming bilingual) is a core component of immigrant children’s adaptive development with many ensuing benefits. HL proficiency has been associated with harmonious family relationships, long-term emotional well-being, stronger identity construction, and psychological adjustment [3–6]. Moreover, combined proficiency in both languages has been linked to abilities that foster academic achievement and support social navigation of youth’s interpersonal worlds [7].

Navigating multiple languages may be demanding. When bilingual developmental difficulties arise, they pose complex challenges for health professionals, educators, and families alike. While numerous standardized tools are available to assess impairment and guide interventions in monolingual contexts, the situation is more complex in cross-cultural and bilingual contexts. The wide variability of language acquisition makes it difficult to distinguish language disorders from language differences, and additional migration-related risks such as discrimination and acculturative stress may further compound the challenges [8],[9]. As highlighted by Stow and Dodd, equitable clinical services require culturally and linguistically appropriate tools and practices to prevent intervention inequities [10]. Identifying factors that may favor or impede bilingual development is therefore essential to help educators, policymakers and health professionals address the specific needs of migrant bilingual children, and optimize interventions to support their language development [11,12].

Reinforce the theoretical and conceptual approach of the subject matter.

Response: Thank you for your suggestion. We have added the text below to explain our theoretical and conceptual approach.

Our research is grounded in the transcultural clinical approach, the theoretical and methodological foundations of which were built on Devereux’s complementarist paradigm, integrating complementary frames of reference, historically, anthropology and psychoanalysis [34,35]. By transcultural, we refer to the dynamic process of cultural transformation with migration, which combines the loss of existing cultural elements, the borrowing and reappropriation of external cultural traits, and the creation of new cultural forms emerging from cultural interaction [36]. This framework has been further operationalized in transcultural psychotherapy (a second-line method drawing on systemic and psychoanalytic family therapy, narrative therapy, and cultural mediation) that has been developed for migrant patients when standard care proves insufficient [37]. Building on this tradition, our clinical work on language and learning disorders with second-generation migrant children has extended the complementarist paradigm. This work suggests that bilingual acquisition must be understood ecologically—not only from a cognitive viewpoint but also from traditional psychological approaches (parents’ subjective experiences) and socio-cultural perspectives (anthropology, sociolinguistics, etc.) [38]. This original approach resonates to a certain extent with the theoretical underpinnings of several ecological and integrative models, such as the Integrative Risk and Resilience Model of Adaptation in Immigrant-origin Children [39,40] or the concept of familism [30].

Explain in depth the reason for using mixed methods and their advantages to approach the population of interest.

Response: Thank you for your question.

We wish to clarify that this study does not employ a mixed-methods approach in the strict sense, as it does not include a qualitative analysis of the corpus. Nevertheless, the study design was inspired by the mixed-methods framework. We chose this approach as the most appropriate for capturing subjective phenomena, particularly parental representations of family transmission. This approach allowed parents to voice their feelings and opinions—to explore what they think they do and how they think about their roles in cultural and linguistic transmission. Qualitative data enabled access to these representations in a more nuanced and open-ended way than standardized questionnaires, which, to the best of our knowledge, did not exist for this purpose when we planned this study.

The second phase involved quantifying qualitative data, allowing us to test correlation hypotheses based on a sufficiently large sample size. This analytic strategy aligned with our theoretical framework, which considers migration as a transformative experience that shapes family relationships—both locally and transnationally—and involves a process of transculturation, through which parents integrate elements of the host culture while maintaining or modifying aspects of their heritage.

We constructed the questionnaire in accordance with this paradigm. Explanatory variables were defined through thematically grouped questions, each designed to explore a specific aspect of parental perception. A scoring system was developed to assess the intensity of investment in familial relationships, cultural affiliations, and perceived life changes since migration.

We have added the following paragraph to the manuscript to explain the reason for using a method inspired by mixed methods:

This study used a cross-sectional design combining standardized language assessments with an innovative approach to capture parental transcultural perceptions. We drew on procedures inspired by mixed-methods designs. This two-step process involved conducting semi-structured interviews with parents and then quantitizing their narrative responses (i.e., transforming them into numerical scores [44]), thereby generating reproducible variables for statistical analysis. The similar strategies applied in health and social sciences to quantify familial relationships [45] inspired the approach used here. The rationale for this approach was twofold. First, interviews allow parents to voice their feelings and opinions, providing access to their subjective experiences of complex transcultural domains in a more open-ended manner than would be possible with a standardized questionnaire; moreover, when we began this study, no standardized questionnaire existed on this subject, nor has any been published since. Second, a quantitative step is required to test associations between transcultural parental factors and children’s receptive and expressive skills in HL and ML. By grounding the construction of quantitative variables in qualitative parental accounts, our design aimed to operationalize complex transcultural constructs and to explore their relationships with children’s bilingual development in ways that reflect the complexity of their language pathways in migration. This strategy aligned with our theoretical framework of complementary perspectives and allowed us to capitalize on the strengths of each tradition—qualitative data offering depth and contextual richness, and quantitative modeling providing statistical robustness and generalizable estimates—to enhance the interpretability of our findings.

Expand on the profile and experience of those who applied the measurement instruments.

Response:

We thank you for your question. The following text clarifies who administered the measurement tools used in the study:

These tools were administered by 7 professionals: 5 clinical psychologists with 10 years of specialized experience in individual, group, and transcultural therapy; a psycholinguist specialized in bilingualism and language assessment with 20 years of experience; and a child/adolescent psychiatrist with 15 years of specialization in language assessment in bilingual contexts and in transcultural work with migrant children and their families.

This group is part of a transcultural research team that has been working in the Paris region for over thirty years, developing research on language and mental health among migrant children to better understand their specific vulnerabilities and how to provide them with effective care.

The problem addressed by the study is relevant to improving the functioning of a multicultural society. It would be desirable to delve into children’s health education and literacy as part of their socialization process.

Response: Thank you for your question. We have added a paragraph at the end of the discussion addressing the implications of our research.

Practical implications : Our results point to several practical implications. First, we suggest that parental transcultural perceptions—how parents value migration-related changes, extended family relationships, and intergenerational transmission—play a role in children’s HL acquisition. Extrapolating these results, we propose that in clinical and educational settings, questions about children’s language use should not be limited to assessing developmental milestones, but could also serve as entry points into understanding identity formation and family relationships. Systematically documenting such perceptions could help identify children at risk of HL erosion and guide more tailored support. Acknowledging transcultural factors that support language development may help families and educators sustain and validate home-based practices as well as contribute to reducing the stigma often attached to minority languages.

Our findings also have implications for children’s health education: supporting HL alongside ML may strengthen health literacy by enabling children to navigate multiple cultural contexts, communicate effectively with both family and healthcare providers, and integrate health knowledge into daily life. Ultimately, rather than viewing multilingualism as a barrier, this approach reframes it as a resource to be nurtured—for the benefit of children, families, and society.

It is a good study, and I suggest improving the manuscript before sending it to other reviewers. 1) to delve into the theories behind the concepts used. Concepts such as inherited language, extended family relationships and transculturality.

Response: Thank you very much for your interest and your encouraging review. We appreciate your suggestion and have now added details on inherited language, extended family relationships, and transculturality in the introduction.

Within this framework, the term heritage language (HL) is today conceptualized as a nondominant language in a host society that holds personal or cultural significance for the individual (e.g., through family or historical ties), regardless of actual proficiency [41]. Unlike alternative labels such as L1 or mother tongue, the concept of HL shifts the focus from linguistic dimension to a symbolic one; HLs are conceptualized as embodiments of community traditions and identities, with each HL a critical medium for transmitting collective memories, values, and cultural practices [42]. The concept of heritage explicitly embeds an intergenerational perspective, linking language to family transmission across time. Pragmatically, HL proficiency may be necessary to sustain communication with relatives, particularly those grandparents who do not master their grandchildren’s ML. Relationships with extended family members also broaden the contexts in which the HL is used, enriching children’s HL environment both quantitatively and qualitatively. Beyond these practical reasons, qualitative studies have also shed light on the HL’s emotional aspects. For instance, some authors, finding that families express a strong emotional connection to the HL, have proposed that HL is valued for its symbolic maintenance of family bonds and cultural identity [43]. Affective ties with extended kin [30] may lead migrants to make a significant effort to provide crucial support for HL use and for its transmission to their children.

By transcultural, we refer to the dynamic process of cultural transformation with migration, which combines the loss of existing cultural elements, the borrowing and reappropriation of external cultural traits, and the creation of new cultural forms emerging from cultural interaction [36].

I suggest to the authors that in their methods they should make explicit the justification of: why is it sufficient to study population with three languages -Arabic, Tamil and Soninke-, is it sufficient to consider them representative of transculturality?

Response: Thank you for your question. We have added a paragraph clarifying how the three contexts provide a rationale for adopting a transcultural approach.

The rationale for the selection of this population was to maximize cultural diversity through a strategic yet pragmatic selectio

---

## [Editor Report · Decision Letter 1]

30 Sep 2025

How migrants’ transcultural perceptions shape their children’s bilingual language development: Insights from a cross-sectional multicultural study

PONE-D-24-60523R1

Dear Dr. Marie Rose Moro,

We’re pleased to inform you that your manuscript has been judged scientifically suitable for publication and will be formally accepted for publication once it meets all outstanding technical requirements.

Kind regards,

Doris Verónica Ortega-Altamirano, PhD

Academic Editor

PLOS ONE

Additional Editor Comments (optional):

The manuscript was modified according to the suggestions of the reviewers and the editor.

The data collection period runs from 2011 to 2014. What is the rationale for releasing data that is 10 years old?

It seems to me that the manuscript contains a spelling error: it says interrate agreement and should have hyphens: inter-rate-agreement.

The data collection period runs from 2011 to 2014.
---

## [Editor Report · Acceptance letter]

PONE-D-24-60523R1

PLOS ONE

Dear Dr. Moro,

I'm pleased to inform you that your manuscript has been deemed suitable for publication in PLOS ONE. Congratulations! Your manuscript is now being handed over to our production team.

Kind regards,

on behalf of

Dr. Doris Verónica Ortega-Altamirano

Academic Editor

PLOS ONE